# Partitioning of Persistent Organic Pollutants between Adipose Tissue and Serum in Human Studies

**DOI:** 10.3390/toxics11010041

**Published:** 2022-12-31

**Authors:** Meg-Anne Moriceau, German Cano-Sancho, MinJi Kim, Xavier Coumoul, Claude Emond, Juan-Pedro Arrebola, Jean-Philippe Antignac, Karine Audouze, Christophe Rousselle

**Affiliations:** 1Oniris, INRAE, LABERCA, 44300 Nantes, France; 2INSERM UMR-S 1124, Université Paris Sorbonne Nord, 93017 Bobigny, France; 3INSERM UMR-S 1124, Université Paris Cité, 45 rue des Saints-Pères, 75006 Paris, France; 4School of Public Health, Department of Environmental and Occupational Health, University of Montreal, Montreal, QC H3C 3J7, Canada; 5Department of Preventive Medicine and Public Health, Universidad de Granada, Campus de Cartuja s/n, 18071 Granada, Spain; 6Instituto de Investigación Biosanitaria (ibs.GRANADA), Avda. de Madrid, 15. Pabellón de Consultas Externas 2, 2a Planta, 18012 Granada, Spain; 7Consortium for Biomedical Research in Epidemiology and Public Health (CIBERESP), Instituto de Salud Carlos III, 28029 Madrid, Spain; 8ANSES, European and International Affairs Department, 14 rue Pierre et Marie Curie, 94701 Maisons-Alfort, France

**Keywords:** adipose tissue, biomonitoring, meta-regression, partition coefficients, persistent organic pollutants

## Abstract

Blood is the most widely used matrix for biomonitoring of persistent organic pollutants (POPs). It is assumed that POPs are homogenously distributed within body lipids at steady state; however, the variability underlying the partitioning of POPs between fat compartments is poorly understood. Hence, the objective of this study was to review the state of the science about the relationships of POPs between adipose tissue and serum in humans. We conducted a narrative literature review of human observational studies reporting concentrations of POPs in paired samples of adipose tissue with other lipid-based compartments (e.g., serum lipids). The searches were conducted in SCOPUS and PUBMED. A meta-regression was performed to identify factors responsible for variability. All included studies reported high variability in the partition coefficients of POPs, mainly between adipose tissue and serum. The number of halogen atoms was the physicochemical variable most strongly and positively associated with the partition ratios, whereas body mass index was the main biological factor positively and significantly associated. To conclude, although this study provides a better understanding of partitioning of POPs to refine physiologically based pharmacokinetic and epidemiological models, further research is still needed to determine other key factors involved in the partitioning of POPs.

## 1. Introduction

Persistent organic pollutants (POPs) are chemicals from a large class of substances heavily produced and released in the environment during the last century, peaking in the 1970s and switching to declining trends when regulations and bans began to be enforced [1]. These chemicals attracted international attention because of their high persistence in the environment and capacity to be transported over long distances and absorbed in the fat tissues of living organisms, prompting their magnification along trophic chains [2,3]. Additionally, many POPs may elicit adverse health effects on animals and humans, motivating global regulatory frameworks such as the Stockholm Convention, adopted in 2004, intended to reduce and monitor their worldwide production. Despite regulatory efforts, substantial concentrations of POPs can still be found in ecosystems, or in fatty tissues and specimens from animals and humans across the globe, decades after the banning period [1,4,5]. The vast family of POPs includes legacy chemicals intentionally produced for agricultural applications, like pesticides (e.g., organochlorinated pesticides), or for industrial uses, including polychlorinated biphenyls (PCBs), which can also be unintentionally released through thermal processes [6]. In addition, other classes, like polybrominated diphenyl ethers (PBDE) used as additive flame retardants, are among the substances of emerging concern [2]. Some POPs have shown the capacity for interacting with and disrupting a variety of molecular pathways, triggering pathogenic processes involved in human health, including impaired reproduction [7,8], carcinogenesis [9,10] or metabolic dysregulation [11,12], among others. Experimental and epidemiological studies on POPs have mainly relied on the effects of a few individual congeners, whereas the effect of low-dose mixtures of POPs resembling realistic background exposures remains scarcely explored [13]. Computational systems biology modeling also offers a new way of investigating potential toxicological effects of POPs [14].

Globally, POPs are characterized by their hydrophobic properties, dissolving in lipid-based foodstuffs like fatty fish or butter, favoring diet becoming a major entry pathway in humans [15]. The mechanism of absorption for many hydrophobic pollutants is the same as for dietary lipids, including the lymphatic pathway or the portal vein to the liver, and is determined by the octanol–water partition coefficients (log *K*_ow_) of chemicals [16]. For instance, the absorption of the highly lipophilic pesticide hexachlorobenzene (HCB, log *K_ow_* =5.7) mostly happens in the intestine; it is transported through the lymph to the tissues within chylomicrons, being mainly stored in adipose tissue [17]. POPs stored in adipocytes are mobilized with neutral lipids to systemic circulation upon energy demands and packed within lipoproteins with lipids and other hydrophobic chemicals. Thus, blood is often used as a convenient matrix for biomonitoring purposes, yet adipose tissue is considered the gold-standard matrix for long-term POP exposure assessment [18]. Elimination of POPs is conducted through bile acids into the intestine where they can ultimately be evacuated within feces, whereas they can be reabsorbed within the enterohepatic recycling process of conjugated bile acids [19]. The dynamics of lipids, and thus of POPs, may be impaired through metabolic disorders (e.g., obesity, diabetes), and POPs may disrupt lipid metabolism and energy balance [20]. In addition, the anthropometry of individuals and variations of fat pad volumes during the lifespan may modify the kinetics of POPs and the chemical exposure biomarkers in environmental health studies [21]. 

In the last decade, pharmacokinetic models, and especially physiologically based pharmacokinetic (PBPK) models, have been developed to refine risk assessment of chemicals [22]. PBPK models have also been used in epidemiological studies to refine the estimation of individual exposures at different critical life stages and/or in different target tissues [23]. They allow the simulation of toxicokinetic profiles for different sub-populations exposed to lipophilic pollutants from single-time-point measurements of POP concentrations in different matrices (e.g., serum, milk), potentially highlighting critical periods of vulnerability to the toxicity of POPs [24,25,26]. These models incorporate physiological parameters such as tissue-specific blood flows, metabolic processes, and tissue partitioning. An assumption usually made for those models is a homogeneous distribution of POPs within body lipids at steady state, thus a partitioning between adipose tissue and blood lipids close to the unit [27,28]. The same assumption has also been generalized to the partitioning of POPs between blood and breast milk if the concentrations are reported in lipid units, yet some authors have reported that this approach may be too simplistic [29,30]. Compound-specific physicochemical properties (e.g., log *K*_ow_, molar volume) and individual physiological parameters (e.g., age, body mass index) may influence such assumptions in the case of breast milk [30]. Nonetheless, little attention has been paid to the partitioning of POPs between adipose tissue and blood lipids; hence, further characterization of POPs’ dynamics through lipid compartments is therefore needed to refine the parametrization of PBPK models and right interpretation of blood biomarkers in epidemiological settings. 

Recent publications have previously addressed the interest in adipose tissue for biomonitoring of environmental chemicals, with a special focus on women [31] and also in epidemiological studies [12], highlighting that adipocytes are also the target of POP. To the best of our knowledge, there are no published studies reporting biomonitoring data on POPs in matched lipid-based matrices (i.e., adipose tissues and blood). Another research gap concerns the biological and physicochemical factors influencing the variability in partitioning of POPs among lipid-based tissues reported in individual studies. Thus, the main objective of this study was to review the state of the science of interrelationships in POP concentrations in adipose tissue with other internal lipid-based compartments including blood lipids, excluding breast milk. Hence, we conducted a narrative literature review of primary articles reporting concentrations of POPs in paired samples of adipose tissue with blood (e.g., serum) or other measured internal compartments. For those articles with reported partition coefficients (e.g., adipose tissue: serum) in comparable metrics, a meta-regression was conducted to identify factors responsible for the variability, otherwise a qualitative synthesis is provided.

## 2. Methods

A literature search strategy was developed to identify recent human studies that measured POPs in serum and adipose tissue from the same individuals. Although the review is essentially narrative, we used the principles of systematic review methodology to ensure the transparency and reproducibility of the process and PRISMA guidelines for reporting [32]. Hence, the bibliographic search was carried out using harmonized standards [33] to develop the search protocol. In this regard, the search strategy was developed to reflect the main key elements of the PECO statement (Table 1), including search strings developed to cover the relevant articles in Scopus and PubMed. The search strings detailed in Appendix A were used in Scopus and PUBMED on 05/08/2022. A ‘publication date’ filter was applied from 2011 to July 2021 in order to focus on the most recent studies that use highly sensitive analytical methods. Manual searches were conducted to retrieve studies published before 2011, based on references cited in the selected studies and the expert knowledge of the panel.

To be included, studies had to be conducted in human populations, measuring internal levels of any type of POP in at least one adipose tissue location and serum or another different adipose tissue location. Hence, articles were excluded when concentrations of POPs in human adipose tissue, either subcutaneous adipose tissue or visceral adipose tissue and serum or in different adipose tissue compartments, were not reported. The search was restricted to articles in English and primary studies (review and conference publications were excluded). 

### Synthesis and Meta-Analysis

Studies with comparative metrics (e.g., ratio of POP concentrations in adipose tissue: serum-determined lipid-based weight, p.e. ng/g lipid weight) were pooled in a meta-analysis. The effect sizes were the adipose tissue ratio means and the respective standard deviations. The means and standard deviations were estimated from the median and interquartile range using the Wan method [34] for two studies [35,36]. Meta-estimates were appraised using a random-effect meta-analysis approach implemented with the *metamean* function in the “meta” R package [37,38]. Higgins & Thompson’s *I*^2^ statistic was used to quantify the between-study heterogeneity. The influence of variables on meta-estimates was assessed by linear meta-regression using the restricted maximum likelihood method and implemented with the *metareg* function of the same package. In this regard, individual variables (e.g., age, body mass index), year of collection and physicochemical properties (summarized in Table 2) were assessed in single and multivariate models. A narrative synthesis displaying the study design and main outputs was conducted for the rest of studies excluded from the meta-analysis, i.e., those for which no ratio of POP concentrations in adipose tissue: serum was provided. 

## 3. Results

### 3.1. Study Characteristics

Finally, 249 publications were initially gathered from PubMed and SCOPUS after duplicate removal, among which 212 did not match the selection criteria and thus were excluded based on title and abstract (Figure 1). The remaining 37 references were evaluated based on the full text, of which 18 studies were considered eligible articles. A final list of 19 relevant articles reporting concentrations of POPs in adipose tissue and serum was included, of which 6 provided comparable metrics to conduct a meta-analysis (Table 2). 

The main characteristics of 19 included studies are displayed in Table 3; they were published between 1999 [43] and 2021 [44]. Among these studies, 11 were performed in Europe and 8 outside Europe. It should be noticed that, for most of these studies, the sampling was conducted several years before publication, extending the exposure timeframe by several years. Different families of compounds belonging to POPs were measured depending on the study: the most frequently measured compounds were PCBs, either as individual congeners such as PCB 153, 138, or 180, or as a sum of PCBs, followed by 1,1,1-Trichloro-2,2-bis(4-chlorophenyl)ethane (*p*,*p*’-DDT), 1,1-dichloro-2,2-bis(p-chlorophenyl) ethylene (*p*,*p*’-DDE), HCB, beta hexachlorocyclohexane (β-HCH) and PBDE. Most studies represented a clinical population undergoing surgery exposed to background levels of POPs, but several involved occupational exposures [36] or highly contaminated areas [45]. The study participants provided adipose tissue biopsies for the surgical intervention during caesareans [36], breast cancer [10,43] or hepatocarcinoma interventions [45], for endometriosis [46,47], or for bariatric surgery [48,49]. Hence, the majority of the population was represented by females with a minor presence of males. Mean ages ranged between 29 and 59 years, and BMI was between 21 and 48 kg/m^2^. Subcutaneous adipose tissue (SAT) was the most frequent location for biopsies, yet some other locations were also reported, including visceral adipose tissue (VAT), breast (BAT) and the gluteal region (GAT). A few studies measured concentrations of POPs in two different adipose tissue types/locations simultaneously [49,50,51]. Nine studies reported or provided the adipose tissue: serum ratios, of which one only reported ratios expressing serum concentrations in wet weight [47], three were either in lipid weight or wet weight, and six were in lipid weight.

### 3.2. Partition Coefficients of Persistent Organic Pollutants between Adipose Tissue and Serum

The studies tended to report the ratios considering both metrics on a lipid weight basis, but in some cases (Table 4) reported both lipid weight and wet weight biomarkers [35,36,51,59]. One study chose to present the values of ratios in wet weight [47]. Among PCBs, the congeners 138, 153 and 180 were the most reported ones in the literature because of their abundance (eight studies). In most cases, the congeners showed adipose tissue: serum ratios between 1.5 and 2; however, some exceptions were found. For instance, a few studies showed mean ratios of PCB 153 close to 5 [35,44], whereas others were close to 0.7–0.9 for the same congener [35,59]. In turn, the metabolite of the insecticide DDT, *p*,*p*’-DDE, was the organochlorinated pesticide (OCP) most widely reported in the six studies. As previously described for PCBs, the mean levels of ratios of OCP were mostly in the range 1.5–2, with several exceptions. The adipose tissue: serum ratios of brominated flame retardants (BFRs) were reported in three studies. Two French studies tended to show consistent values for most PBDEs [10,46], showing mean values in the range between two and three for most congeners with the exception of BDE209, which showed the highest values (4.6–8.0). Conversely, in one study conducted in an e-waste facility area with expectant women, the ratios tended to be below the unit for most congeners (0.4–1.1) [36]. In summary, for most studies, the reported adipose tissue: serum ratios displayed a similar pattern, with a 1.5-3-fold increased concentration of POPs in the adipose tissue, considering the measures based on lipid weight.

### 3.3. Physicochemical Determinants of the Variability in Adipose Tissue: Serum Partition Coefficients 

In order to gain insight into the physicochemical factors potentially affecting the variability in adipose tissue: serum ratios, we conducted a meta-regression based on available summary partitioning data reported in the selected studies. The meta-regression slopes, graphically depicted by bubble plots, are displayed in Figure 2A–D. The number of halogens was the variable most strongly associated with adipose tissue: serum ratios (β 0.25 95% CI (0.17–0.34), *p* = 0.05 (Figure 2B)). Log *K*_ow_, molar weight (M_w_) and molar volume showed null or mild associations with the ratios (Figure 2A,C,D).

### 3.4. Biological/Individual/Instrumental Determinants of the Variability in Adipose Tissue: Serum Partition Coefficients

Among individual variables, BMI was the most strongly associated with the adipose tissue: serum ratios for all POPs combined (Figure 3A), β 0.26 95% CI (0.17–0.34), *p* < 0.001, whereas age showed a more flattened slope than BMI, yet was statistically significant, β 0.02 95% CI (0.00–0.03), *p* = 0.03 (Figure 3B). Due to the close relationship between age and BMI, we also considered both variables in the same meta-regression model, suggesting a potential confounding effect of age, resulting in null associations for age with a minor impact on BMI coefficient. The year of sample collection (Figure 3C) also showed a positive association with the ratios β 0.26 95% CI (0.17–0.34), *p* < 0.001. For the congeners with at least three studies reporting the adipose tissue: serum ratios and BMI, we conducted a detailed meta-regression for individual congeners of PCBs and BDEs (Figure 4A–F). These stratified results supported the existence of congener-specific relationships between the ratios and BMI. For instance, BDE 209 was associated with BMI with the steepest slope, whereas PCB180 showed the most flattened one (Figure 4).

### 3.5. Variability in Levels of POPs between Adipose Tissue Types/Location

Seven studies analysed POPs in different adipose tissue locations (mainly visceral and subcutaneous) matched from the same individuals, including obese populations and women with endometriosis [46,48,49,50,56,58,59]. Globally, concentrations of organochlorinated and brominated chemicals analysed in paired VAT and SAT matrices showed high correlations, for instance with R Pearson correlation >0.9 (*p* < 0.001) for all POPs from France [46] or R^2^ 0.7–0.6 (*p* < 0.0001) for PCBs from Belgium [49]. Congener-specific and strong sex-specific differences were noticed for the correlation slopes, showing lower correlations of POPs between VAT and SAT among females [49]. Noteworthy, two studies reported high correlations for individual POPs or ∑POPs between VAT and SAT locations but statistically different concentrations, with higher concentrations of individual POPs or ∑POPs in VAT than SAT [56,58]. Concentrations of organochlorinated pesticides (*p*,*p*’-DDE, *p*,*p*’-DDT, β-HCH) and PCBs (153 and 180) were also highly correlated between breast and gluteal fat pads (rho > 0.7, *p* < 0.001), showing comparable concentration levels [59]. The relationships between POPs in human foetal adipose tissue and maternal serum or placenta have been addressed by one study, showing in general high correlations for all PCBs and pesticides analysed (HCB, β-HCH, *p*,*p*’-DDE and transnonachlor) [55]. The authors reported sex-specific patterns of tissue accumulation, reporting higher tissue ratios among male foetuses than among females and lower among those with placental insufficiency. 

## 4. Discussion

To the best of our knowledge, this is the first study attempting to summarize the available scientific evidence on partitioning of POPs between adipose tissue and serum from human studies. We have also aimed at characterizing and gaining insight into the determining factors of variability for partitions, as was already done for partitioning between serum and breast milk [30]. Data were retrieved mostly from studies conducted with non-occupationally exposed populations, reporting matched samples on lipid-based compartments, thus illustrating chronic low-dose exposure to POPs. Adipose tissue is acknowledged to be the gold-standard matrix for biomonitoring internal POP levels, yet blood (lipids) has been used as a surrogate for many years, assuming that blood POPs reflect their adipose tissue levels in equilibrated states [60]. Consequently, blood concentrations of POPs are extensively used for biomonitoring and epidemiological research. The widespread presence of low doses of POPs in internal biological matrices urges a comprehensive understanding of their kinetics for accurate use of exposure biomarkers in biomonitoring and epidemiological studies.

### 4.1. Overall Variability in Adipose Tissue: Serum Partition Coefficients among POPs

All individual studies included in this review reported high variability in the partition coefficients of POPs between adipose tissue and serum, at different levels of magnitude depending on the specific congeners. Whereas it has often been assumed that highly lipophilic chemicals would distribute homogeneously within lipid compartments of the human body, similarly to 2,3,7,8-tetrachlorodibenzo-p-dioxin [60], the studies included in this review illustrate that the mean ranges between 0.4 and 8 in a congener-specific base. Most ratios were above 1, suggesting a higher concentration of POPs in adipose tissue than in blood; nonetheless, in a few cases the ratios were inversed, reflecting the highest concentrations in serum. Among the different physicochemical properties investigated in the meta-regression, the number of halogen atoms was the most strongly and positively associated variable with the partition ratios, at the limit of statistical significance (*p* = 0.05), with the rest being statistically null (Log *K*_ow_, Mw, and molar volume). The bubble plots reveal the strong dispersion of data for BDE209 and the underlying correlation of physicochemical variables (e.g., BDE209 showed the largest physicochemical properties: Log *K*_ow_ = 12.11, number of halogens = 10, molar volume = 428.6 m^3^/mol). In addition, the bubble plots reflect that the linear meta-regression may be strongly influenced by two dominant studies reporting low ratios for BDE209, suggesting a non-monotonic relationship of the ratios with these physicochemical properties. These trends are consistent with the previous literature. For instance, positive associations between log *K*_ow_ and adipose tissue: serum ratios were reported for most POPs with the exception of PCB180 and BDE209, with higher log *K*_ow_ values (8.27 and 12.11, respectively) and a higher number of halogens [36]. Similar trends were reported for the blood/milk partitioning, which decreased among substances with larger log *K*_ow_ (>8), higher molecular weight (>400 for PCB/PCDD/Fs, >700 for PBDEs), and more halogen compounds (>7) [30]. A bioaccumulation factor increase correlated with increased *K_ow_* was reported for lipophilic compounds with *K_ow_* < 7 in wild aquatic species [61]: the bioaccumulation factor would increase for *K_ow_* < 7, then decline for *K_ow_* > 7. A parabolic relationship was also observed between log biomagnification factor and log *K_ow_*, with a peak for *K_ow_* of approximately 8 [62]. Lv et al. (2015) also noted that compounds with similar log *K_ow_* but from different families (PCB vs. PBDE) exhibited different partition ratios, lower for BDEs than PCBs [36]. It was assumed that deposition of PBDE in human adipose tissues was lower than deposition of PCB with similar *K_ow_*; however, these trends were not consistent in other studies [10,46]. 

Globally, these data support that physicochemical parameters, such as number of halogens, may have an impact on POP partitioning between adipose tissue and serum, but likely in a non-monotonic manner and influenced by other parameters. For instance, the tridimensional structure and associated degree of rigidity of the different compounds could also have an impact on the diffusion-limited partition of POPs [30]

### 4.2. Congener-Specific Variability in Adipose Tissue: Serum Partition Coefficients between Studies

Table 4 summarizes the high variability in partition ratios determined for the same congener among different studies. For instance, PCB138’s reported mean ratios ranged from 1.59 to 5.44, a three-fold difference for the same compound. Systematic differences were appreciated between studies, in some cases even switching the direction in the case of BDEs [36]. These differences may be influenced by specific study characteristics, such as timing of sampling, differences in analytical procedures introducing analytical uncertainties for POPs, or the different determinations of lipid content derived from different equations [15].

Among the individual biological factors analyzed, BMI was the main determinant, positively associated with the partition ratios (β 0.26 95% CI (0.17–0.34), *p* < 0.001), while age was associated to a weaker degree and non-associated when both BMI and age were considered in the multivariate meta-regression model. Globally, these findings suggest that the larger the fat mass, the higher the concentration of POPs in adipose tissue relative to blood lipids, independently of age. These findings would empirically support the hypothesis that populations with high adipose fractions may be sequestering more POPs in their adipocytes, preventing their release into circulation and thus protecting other organs [63]. Most studies that reported different types of adipose tissue deposits did not support differences between locations (e.g., VAT vs. SAT), as high or similar concentrations were reported. In any case, the number of studies was very limited to include the adipose tissue location as a co-variable in the meta-regression or to conduct a stratification analysis. One study showed different trends, with high correlations between ∑POP and AT, with the highest concentrations in VAT rather than SAT [56,58]. These differences may be due to the dissimilarities in lipid content [58], differences in blood flow, or in the density of blood vessels and/or differences in metabolism, with VAT being more metabolically active [64,65]. A PBPK model was proposed for γ-HBCD in mice [66], including an additional compartment corresponding to the deep fat fraction of the body. For POPs with log now >4, it was suggested that ratio of the neutral lipid-equivalent content (corresponding to neutral lipids and phospholipids) of adipose tissue and blood was the main determinant of POP partitioning between those tissues [67]: different types of lipids and compositions of non-lipid molecules might play a role in POP dynamics within AT, and interactions between POPs and lipid compounds at the molecular scale should be further investigated [46], especially for biomonitoring with blood. It was also suggested that extracellular and lymphatic circulation might contribute to POP storage [66]. Metabolic status and disease condition may also play a role in POP partitioning [15], but those parameters were not investigated in the meta-regression due to the limited number of studies. The level and the route of exposure could also have an impact on the metabolic disruption of adipocytes, and thus the kinetics of POPs. In this regard, the exposure scenario may also contribute to explain the variabilities of coefficients. Nonetheless, formal statistics were not applicable to compare background exposures with occupational or highly contaminated sites. For instance, one occupational e-waste study [36] showed serum levels of some BFRs 20 times higher than in populations exposed at background levels [10,46], which would explain ratios systematically lower, below the unit.

### 4.3. Limitations of the Review

First, the current review was essentially narrative, even if a robust methodology based on systematic review principles was applied. If our review is not exhaustive, we believe that most studies with AT and blood POP measurements published during the last decade have been included. Because of this criterion, only a limited number of studies were eligible in the meta-analysis. In addition, each individual study involved a modest sample size. In order to strengthen the statistical power, studies with larger sample size will be required. Furthermore, the selected studies explored different sets of congeners, limiting the ability to compare ratios among studies for the same compounds and to conclude about the influence of population biological characteristics. Actually, the studies provided few individual variables that could be integrated in the meta-regression, but other factors may likely impact on the kinetics of POPs and consequently on the ratios (e.g., parity, breastfeeding, physical activity). In addition, we pooled some summary metrics through approximations, because not all studies reported the means and standard deviations. Optimally, statistical analyses combining raw data from individual studies would improve the accuracy of estimates. Considering the invasive character of adipose tissue biopsies collected from the populations with cancer or obesity the data may not be representative of the general population. This review also stressed the need for further studies on dioxins scarcely reported in the literature. Per/polyfluorinated substances (PFAS) are a wide class of chemicals with some congeners listed as POPs in the Stockholm Convention commonly characterized in blood due to their amphiphilic properties; hence, they are not represented in the present review because they are more present in livers than in AT. Nonetheless, some studies have shown that even non-persistent chemicals like bisphenol A or parabens can be found in adipose tissue [68,69]. New approaches based on artificial intelligence, which used text mining to automatically screen large literature sources and databases, could also be considered in order to get as much information as possible. As an example, the AOP-helpFinder tool [70,71], which helps the development of adverse outcome pathways, could be adapted in future meta-analysis studies.

### 4.4. Impacts of Findings

We believe that the findings of this study are of high relevance due to the widespread use of exposure biomarkers of POPs for biomonitoring and epidemiological research. Biomonitoring of chemicals has been progressively integrated as a novel tool within the chemical risk assessment process, notably by using PBPK modelling and the development of biomonitoring equivalents [72,73,74]. PBPK models of POPs are often based on the assumption of a homogeneous distribution of POP in lipid compartments within the body (i.e., that adipose tissue: blood ratios are comparable). Thus, a better understanding of partitioning of POPs between fat compartments will help the refinement of those models and reduce uncertainty within chemical risk assessment and regulatory decision-making. Likewise, in epidemiological research, the extensive use of blood biomarkers of POPs has not been supported by a deep understanding of the complex relationship of POPs in blood and adipose tissue [15]. Nowadays, different statistical methods have been proposed to use the blood biomarkers, addressing the different causal structures depicted when the perturbation of lipid metabolism is in the causal pathway between the exposure and the health outcomes [75]. Although some methods may attenuate the statistical bias, none of the methods using a single-spot blood biomarker may fully recapitulate the complex dynamics of POPs. The problem becomes more complex from the perspective of POP mixtures, where individual congeners may impact the kinetics of the others. Hence, a better understanding of the dynamics of partitions between fat compartments is of key relevance for accurate use and interpretation of surrogates’ measures in environmental health studies. 

## 5. Conclusions

This review supports the fact that partitioning of POPs between fat compartments is highly variable. These findings also show that the partition ratios of POPs between fat compartments systematically depart from the unit, even at equilibrated states, as extensively assumed. BMI and number of halogens were the main determinants of the variability in the adipose tissue: serum ratio of POP among the selected studies. Finally, this study highlights that further research is still needed to better understand the behavior of POPs in the main matrices used for biomonitoring, especially for those scarcely studied congeners. 

## Figures and Tables

**Figure 1 toxics-11-00041-f001:**
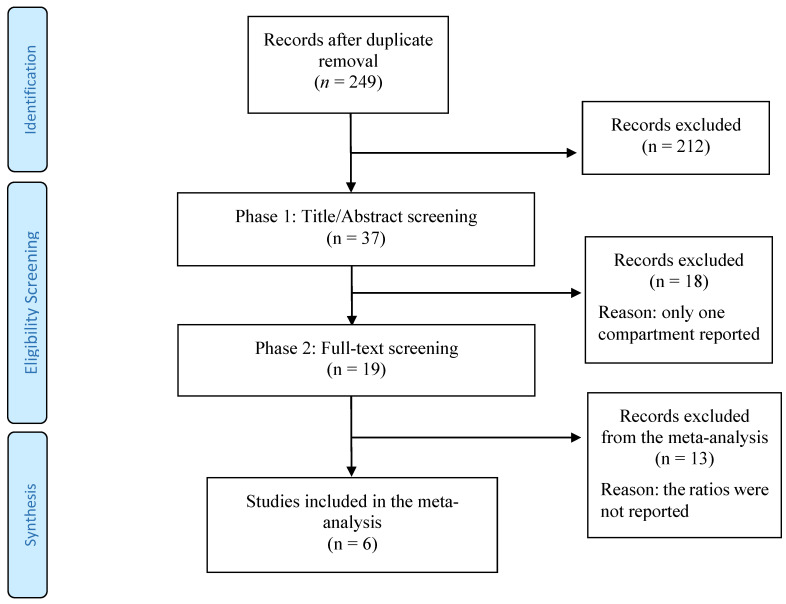
PRISMA flowchart of search strategy and study selection with exclusion criteria.

**Figure 2 toxics-11-00041-f002:**
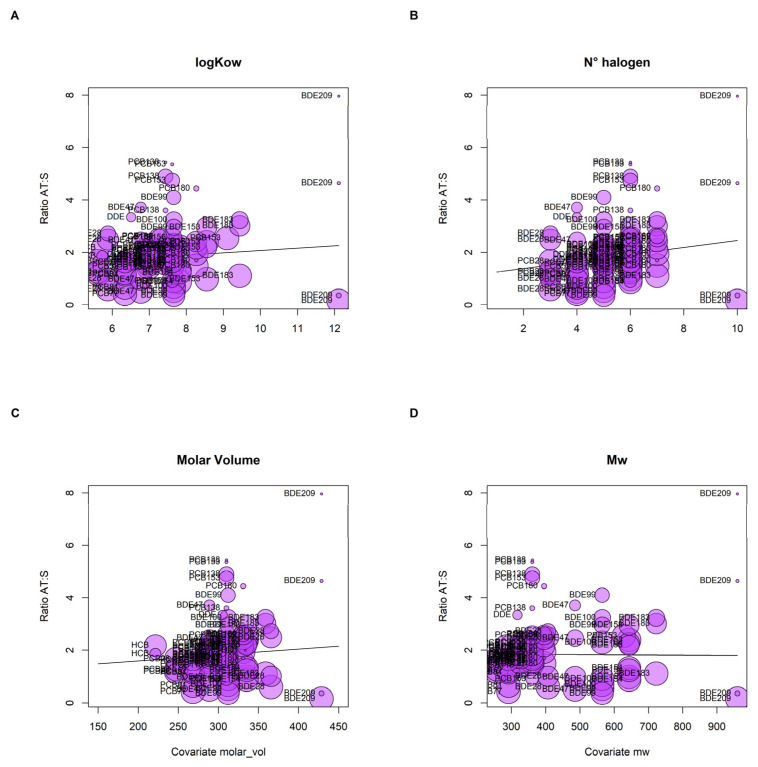
Bubble plots representing the associations between the mean partitioning ratios of persistent organic pollutants between adipose tissue and serum reported in the literature and physicochemical properties, including the partition octanol–water (Log *K*_ow_, (**A**)), number of halogen atoms (**B**), molar volume (**C**) and molar weight (Mw, (**D**)).

**Figure 3 toxics-11-00041-f003:**
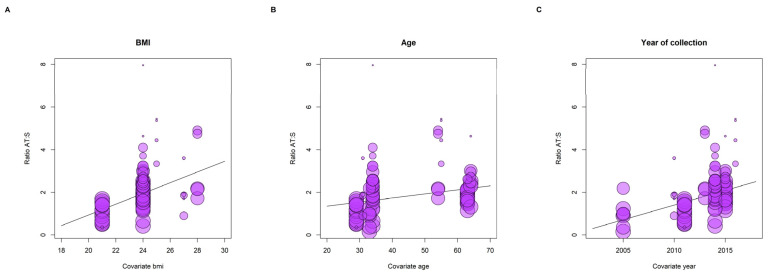
Bubble plots representing the associations between adipose tissue: serum ratios with body mass index (BMI, (**A**)), age (**B**), and year of collection (**C**) from the random effects meta-regression model. The study [52] was excluded from this analysis because the BMI was not reported.

**Figure 4 toxics-11-00041-f004:**
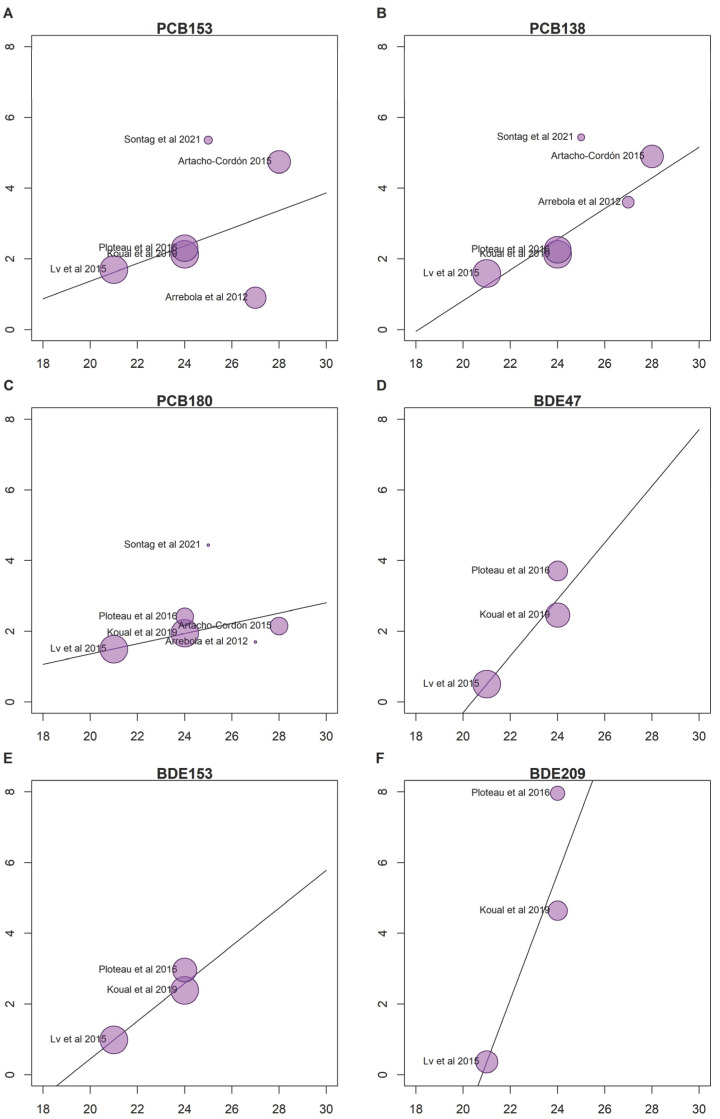
Bubble plots representing the associations between adipose tissue: serum ratios and body mass index (BMI) from the random effects meta-regression model for specific congeners including PCB 153 (**A**), PCB 138 (**B**), PCB 180 (**C**), BDE 47 (**D**), BDE 153 (**E**), and BDE 209 (**F**). The study [52] was excluded from this analysis because the BMI was not reported.

**Table 1 toxics-11-00041-t001:** PECO statement.

PECO	Description
Population	Human populations (including men, women, pregnant women, foetuses exposed in utero, and children exposed).
Exposure	All type of persistent organic pollutants (e.g., PBDEs, PCBs, etc)
Comparator	Not applicable.
Outcome	Simultaneous determination in adipose tissues and serum (or different adipose tissue locations).

**Table 2 toxics-11-00041-t002:** Physicochemical properties of included chemicals retrieved from PUBCHEM database [39] and adapted from previous publications [40,41,42].

Name	CAS N°	Molecular Mass	Log *K*_ow_	Half-Life (Year)	N° Halogen	Molar Volume
Dioxin-like Polychlorinated Biphenyls						
PCB77	32598-13-3	292	6.34	0.1	4	268.2
PCB81	70362-50-4	292	6.34	0.7	4	268.2
PCB126	57465-28-8	326.4	6.98	1.6	5	289.1
PCB169	32774-16-6	360.9	7.62	7.3	6	310
PCB105	32598-14-4	326.4	6.98	2.4	5	289.1
PCB114	74472-37-0	326.4	6.98	10	5	289.1
PCB118	31508-00-6	326.4	6.98	3.8	5	289.1
PCB123	65510-44-3	326.4	6.98	7.4	5	289.1
PCB156	38380-08-4	360.9	7.62	16	6	310
PCB157	69782-90-7	360.9	7.62	18	6	310
PCB167	52663-72-6	360.9	7.62	12	6	310
PCB189	39635-31-9	395.3	7.27	22	7	330.9
Non-dioxin-like Polychlorinated Biphenyls						
PCB28	7012-37-5	257.5	5.62	16	3	247.3
PCB52	35693-99-3	292	6.34	10	4	268.2
PCB101	37680-73-2	326.4	6.98	11	5	289.1
PCB138	35065-28-2	360.9	7.44	27	6	310
PCB153	35065-27-1	360.9	7.62	14.4	6	310
PCB180	35065-29-3	395.3	8.27	11.5	7	330.9
Organochlorinated pesticides						
HCB	118-74-1	284.8	5.73	6	6	221.4
*p*,*p*’-DDE	72-55-9	318	6.51	10	4	305.2
*p*,*p*’-DDT	50-29-3	354.5	6.91	7	5	333.5
Brominated flame retardants						
BDE28	41318-75-6	406.89	5.88	0.94	3	365.5
BDE47	5436-43-1	485.79	6.77	0.37	4	288.8
BDE99	60348-60-9	564.7	7.66	8.2	5	312.1
BDE100	189084-64-8	564.7	7.66	2	5	312.1
BDE153	68631-49-2	643.6	8.55	3.5	6	335.4
BDE154	207122-15-4	643.6	7.82	3.3	6	335.4
BDE183	207122-16-5	722.5	9.44	0.25	7	358.7
BDE209	1163-19-5	959.2	12.11	0.04	10	428.6
PBB153	59080-40-9	627.6	9.1	6.2	6	335.4

**Table 3 toxics-11-00041-t003:** Summary of study characteristics.

Reference	Country	n	Condition	Gender	Age ^a^	BMI ^a^	Year Collection	Adipose Tissue	Serum		Ratio ^c^	Meta ^d^	Dioxin	PCB	OCP	BFR
					(Years)	(kg/m^2^)			Fasting	Units						
[52]	France	86	Caesarean	F	33 (20–46)	NA	2004–2006	SAT	Yes	LW						25
[35]	Spain	103	NA	F	54 (SD:12)	27	2012–2014	BAT	Yes	WW	X			3	2	
[35]	Spain	103	NA	F	54 (SD:12)	27	2012–2014	BAT	Yes	LW	X	X		3	2	
[51]	Bolivia	112	NA	M/F+	31 (18–70)	27	2010	VAT	NA	WW	X			3	3	
[51]	Bolivia	112	NA	M/F+	31 (18–70)	27	2010	VAT	NA	LW	X	X		3	3	
[53]	Spain	387	NA	M/F	52 ^b^ (37–63)	27	2003–2004	SAT	No	LW					1	
[54]	Italy	70	Caesarean	F	33 ^b^ (SD:4)	24	2006	SAT	NA	LW				30	3	
[55]	Sweden	20	Stillbirths	F	25–40	21–35	2015–2016	FAT	NA	WW	X			10	9	3
[50]	Qatar	34	IR	M/F	32 ^b^ (SD: 10)	45	NA	SAT, VAT	No blood							28
[48]	France	42	Obesity/H	M/F	44/40	48/22	2006–2008	SAT(VAT)	NA	W/LW			17	18		
[56]	Korea	50	T2D/N-T2D	M/F	66/62 (8/11)	23 (3)	NA	SAT,VAT	No blood					19	13	
[10]	France	48	BC	F	64	24	2015	BAT	No	LW	X	X		18		8
[36]	China	37–55	Caesarean	F	29	21	2011	SAT	Yes	LW	X	X		6		8
[36]	China	37–55	Caesarean	F	29	21	2011	SAT	Yes	WW	X			6		8
[43]	Mexico	198	BC/H	F	40 (19–78)	NA	1994–1996	BAT	No	LW				1		
[49]	Belgium	52	Obesity	M/F	40 (18–58)	42	2010–2012	SAT, VAT	No blood					28	2	
[57]	Belgium	101	Infertile	F	32 (24–42)		1996–1998	SAT		LW				7	7	
[58]	Portugal	189	Obesity	M/F+	43 (19–65)	45 (5)	2010–2011	SAT, VAT	No blood				1		11	
[46]	France	67	ENDO/H	M+/F	34 (48–80)	24	2013–2015	VAT, SAT	NA	LW	X	X		18		8
[47]	US	473	ENDO/H	F	33 (IQR:11)	26(IQR:10)	2007–2009	VAT	No	WW	X			13	5	5
[59]	India	29	BC	F	24–65	NA	1996–1997	BAT	Yes	WW	X			2	3	
[59]	India	29	BC	F	24–65	NA	1996–1997	GAT	Yes	WW	X			2	3	
[59]	India	29	BC	F	24–65	NA	1996–1997	BAT	Yes	LW	X			2	3	
[59]	India	29	BC	F	24–65	NA	1996–1997	GAT	Yes	LW	X			2	3	
[44]	Australia	32	NA	M/F	55 (22–89)	25	2016	SAT	Yes	LW	X			3		
[45]	Italy	59	HCC	M+/F	69 (48–80)	NA	NA	VAT	Yes	LW	X	X		7		

^a^ Mean age or body mass index and range, standard deviation (SD) or interquartile range (IQR), ^b^ geometric mean or median, ^c^ identifier of studies that reported ratios, ^d^ identifier of studies included in the meta-regression. Abbreviations: BAT, breast adipose tissue; BC, breast cancer; BFR, brominated flame retardants; BMI, body mass index; FAT, fetal adipose tissue; GAT, gluteal adipose tissue; F, female; F+, overrepresentation of female; H, healthy subjects; HCC, hepatocellular carcinoma; IQR, interquartile range; IR, insulin resistance; LW, lipid weight; M, male; NA, not available; OCP, organochlorinated pesticides; PCBs, polychlorinated biphenyls; R, identifier for studies reporting calculated partition ratios; SAT, subcutaneous adipose tissue; SD, standard deviation; VAT, visceral adipose tissue; WW, wet weight.

**Table 4 toxics-11-00041-t004:** Congener-specific summary of mean coefficient partitions of persistent organic pollutants between adipose tissue and serum, both normalized on the lipid weight basis.

Chemical	[52]	[51]	[35]	[10]	[43]	[36]	[46]	[44]	[59] BAT	[59] GAT	[45]
PCB101				1.8			1.5				
PCB105				2.1			2.1				
PCB114				2.1			2.3				
PCB118				1.8		1.4	1.8				1.2
PCB123				1.6			1.9				
PCB126				1.7			1.7				
PCB138		3.6	4.9	2.1		1.6	2.3	5.4			2.0
PCB153		0.9	4.7	2.1		1.7	2.3	5.4	0.7	0.7	2.0
PCB156				2.1			2.6				1.7
PCB157				1.6			1.8				
PCB167				1.5			1.7				
PCB169				1.8			2.0				
PCB169											2.2
PCB180		1.7	2.2	2.0		1.5	2.4	4.4	0.8	0.8	2
PCB189				2.6			2.6				
PCB194											2.1
PCB28				1.7		1.2	1.3				
PCB52				1.5			1.2				
PCB77				1.6			0.4				
PCB81				1.2			0.7				
PCB99						1.4					
*p,p*’-DDE		1.9	1.7		4.2			3.4	0.7	0.9	
*p,p*’-DDT		2							1.1	1.5	
HCB		1.9	2.2								
β-HCH									1.1	1.6	
BDE100	0.91			2.3		0.7	3.2				
BDE153	2.12			2.4		1.0	3.0				
BDE154	1.24			1.3		0.9	2.3				
BDE183				3.0		1.1	3.2				
BDE209	0.16			4.6		0.4	7.9				
BDE28	1.00			2.7		0.6	2.5				
BDE47	0.97			2.5		0.5	3.7				
BDE99	0.36			3.0		0.5	4.1				
PBB153				2.5							

Abbreviations: BAT, brown adipose tissue; BDE, bromodiphenyl ether; β-HCH, β-hexachlorocyclohexane; GAT, gluteal adipose tissue; HCB, hexachlorobenzene; PCB, polychlorinated biphenyl; PBB153, 2,2′,4,4′,5,5′-hexabromobiphenyl; *p*,*p*’-DDT, 1,1,1-Trichloro-2,2-bis(4-chlorophenyl)ethane; *p*,*p*’-DDE, 1,1-dichloro-2,2-bis(p-chlorophenyl) ethylene.

## Data Availability

Not applicable.

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
