# Peer review of "Partitioning of Persistent Organic Pollutants between Adipose Tissue and Serum in Human Studies"

_toxics, 2022, doi:10.3390/toxics11010041_

Round 1
Reviewer 1 Report
Dear Editor,
Thank you for inviting me to review the manuscript entitled “Determinants of partition coefficients adipose tissue: serum variability of persistent organic pollutants: a meta-analysis of human studies”. I have reviewed the manuscript with keen interest and in detail. While the topic is interesting, several issues have been identified, especially in methodology, that should be addressed and specified by the authors satisfactorily. my specific comments are outlined below:
Title:
1. The title is long and should be shortened.
Abstract
2. The introductory sentences are long (lines 23-27) and should be shortened.
- The description of the methodology should be improved.
- The conclusion section of the abstract could be more informative.
- Keywords should be re-ordered alphabetically.
Introduction
6. Please expand the literature review by emphasizing the existing knowledge gaps related to the topic.
7. Please develop your introduction further by highlighting the significance and novelty of your work.
Method
8. Overall, this section should be substantially improved.
9. The authors should explain the search strategy (based on the PECOS format) in detail and present it as supplementary material not in the main text. This also applies to keywords presented in the main text.
10. The authors missed important bibliographic databases such as Embase, Medline, and Web of Science. It is very important to review all the potential resources to retrieve as many relevant references as possible.
- What was your search strategy for gray literature?
12. L 136-141: The authors mentioned that they considered papers that were published between 2011-2021 and passed those published before this period but later, they mentioned they have included some articles published before 2011 in a second step as they were considered reliable and of interest for the review. The information provided is quite confusing. How did you conclude that papers published before 2011 were not of sufficient quality when you did not include them in your search strategy? In addition, how you would find relevant papers published before 2011? Overall, the provided information is quite confusing, and the search strategy does not sound clear and robust.
13. How did you assess the risk of bias of the included studies?
14. The authors should provide more information on the quantitative synthesize of the findings (meta-analysis).
15. How heterogeneity between studies was assessed?
16. L 160: “Qualitative synthesis was conducted for the rest of studies excluded from the meta-analysis”. How did you perform qualitative synthesis? Which method did you use? It should be explained in detail.
Results
17. The authors should add the reasons for excluding studies to figure 1.
Conclusion
18. Based on the results obtained in the present study, what are your suggestions for future studies? Please add your suggestions at the end of the discussion or in the conclusion section.
19. The manuscript should be proofread carefully as typo errors and grammatical mistakes were found in the text.
Reviewer 2 Report
This paper conducted a meta-analysis of persistent organic pollutants (POPs) in adipose tissue and serum in human studies and identified factors responsible of variability between adipose tissue and serum. The detailed comments are listed below:
Line 165-171
The results may be doubted by the present method. References maybe better searched by Web of Science. More relevant articles are needed to better understand the relationship between adipose tissue and serum
Line 278-280
Are concentrations of individual POPs and/or all POPs in VAT higher than SAT?
“Highest” should “higher”
Up to now, 30 chemicals have been added into the list of POPs under the Stockholm convention. More POPs should be included in this review.
Please write a section of conclusion about this review.
Round 2
Reviewer 1 Report
The authors addressed my comments in the revisions. I would recommend the acceptance of the revised manuscript.